# Bias Amplification in Image Classification

## Abstract

Recent research suggests that predictions made by machine-learning models can amplify biases present in the training data. Mitigating such bias amplification requires a deep understanding of the mechanics in modern machine learning that give rise to that amplification. We perform the first systematic, controlled study into when and how bias amplification occurs. To enable this study, we design a simple image-classification problem in which we can tightly control (synthetic) biases. Our study of this problem reveals that the strength of bias amplification is correlated to measures such as model accuracy, model capacity, and amount of training data. We also find that bias amplification can vary greatly during training. Finally, we find that bias amplification may depend on the difficulty of the classification task relative to the difficulty of recognizing group membership: bias amplification appears to occur primarily when it is easier to recognize group membership than class membership. Our results suggest best practices for training machine-learning models that we hope will help pave the way for the development of better mitigation strategies.

## 1 Introduction

Several recent studies have presented results suggesting that, beyond reproducing biases present in the training data, machine-learning models can *amplify* such biases as well [11, 34, 38]. Bias amplification is concerning as it can foster the proliferation of undesired stereotypes [9, 32, 38, 37] or lead to unjustifiable differences in model accuracy between subgroups of users [5, 8].

The existence of bias amplification suggests that machine-learning models are not always doing what we expect them to do: *viz.*, make predictions according to the statistics present in their training data. Although several studies have proposed measures for the severity of bias amplification [11, 25, 34, 38], this question of when and why bias amplification occurs remains largely unanswered.

We present a systematic, controlled study of bias amplification. We design a simple image-classification task that facilitates tight control of synthetic biases. In line with prior work [11, 34, 38], we find that models trained for this classification task, indeed, amplify biases present in their training data. We use the ability to control biases to study key research questions (RQs) that increase our understanding of bias amplification:

- *RQ1*: How does bias amplification vary as the bias in the data varies?
- *RQ2*: How does bias amplification vary as a function of model capacity?
- *RQ3*: How does bias amplification vary during model training?
- *RQ4*: How does bias amplification vary as a function of the relative difficulty of recognizing class membership versus recognizing group membership?

We observe that bias amplification tends to increase with bias in the training set in many of our experiments. We find that bias amplification varies with model capacity: models with more parameters

and/or less regularization can amplify biases, but models with too few parameters and/or too much regularization can amplify biases even more. Bias amplification also greatly varies with training set size: models trained on very small or very large training sets appear to amplify biases less. We observe that the degree of bias amplification can vary greatly during model training. In many of our experiments, we find that the behavior of bias amplification depends on the difficulty of the classification task relative to the difficulty of group membership recognition.

The results of our study provide intuitions for when bias amplification occurs and why. They suggest some best practices that may help reduce bias mitigation in real-world machine-learning models, such as careful cross-validation of hyperparameters related to model capacity, regularization, and training duration to substantially reduce bias amplification of the final model. Collecting more training data may reduce bias amplification as well. We hope that our study helps pave the way for the development and adoption of mitigation strategies for bias amplification in common computer vision tasks.

## 2 Experiments

We design an image-classification task in which each image has both a class and a group, and in which we can introduce synthetic biases by altering the group assignment of images.

### 2.1 Experimental setup

**Classification task.** We perform image-classification experiments on three image datasets: (1) the Fashion MNIST dataset [36], (2) the CIFAR-10 dataset [21], and (3) the CIFAR-100 dataset [21]. Because our analyses are easier to perform with binary classification problems, we convert the datasets to have binary labels by randomly selecting half of the classes to be positive and the other half to be negative. All our models are residual networks [16] that we train according to the procedures discussed in Appendix A.1.1. To mitigate the effect of a particular random class assignment, we average the test accuracy and bias amplification values over 20 random assignments of the original classes to the binary classes and include $95\%$ confidence intervals.

**Group membership.** We consider a classification model to be *biased* if it predicts a particular *class* at a disproportionate rate for examples from a particular *group*. However, rather than using real-world groups we choose to establish synthetic groups for our experiments. This reduces the noise in our measurements and allows us to perform additional root-cause investigations of bias amplification that may pose ethical or technical challenges with real-world groups, such as determining the model's ease of predicting the group in the image.

Specifically, we create two groups in our image-classification problems by *inverting* some of the images in a dataset and not inverting others. Using image inversion to create groups has two main advantages over other synthetic methods like color changes or adding random noise: (1) it hardly introduces new visual features into the images that may alter the image-classification problem and (2) it is straightforward for image-recognition models to recognize whether or not an image is inverted.[1] This allows us to tightly control the correlation between classes and groups without introducing causal relations between them. Figure 6 shows examples of inverted and non-inverted images.

**Controlling dataset bias.** For all images corresponding to a single task in the input dataset, we randomly select positively labeled images with rate $1/2 - \epsilon$ and invert them, and we randomly invert negatively labeled images with rate $1/2 + \epsilon$ (we choose $\epsilon \in [0, 1/2]$). This leads to a bias of strength $2\epsilon$ in the dataset: If $\epsilon = 0$, image inversion (*i.e.*, group membership) carries no information on whether the images has a positive or a negative label (*i.e.*, class membership). By contrast, group membership uniquely defines class membership when $\epsilon = 1/2$. Hence, $\epsilon = 0$ corresponds to an *unbiased* dataset in which group membership does not carry information about class membership, $\epsilon = 1/2$ corresponds the a *fully biased* setting in which group membership uniquely determines class membership, and values of $\epsilon \in (0, 1/2)$ correspond to *partly biased* datasets.

---

[1]In preliminary experiments, we found that the test accuracy of a residual network trained to recognize image inversion is $100\%$ on Fashion MNIST images and $96\%$ on CIFAR-100 images.

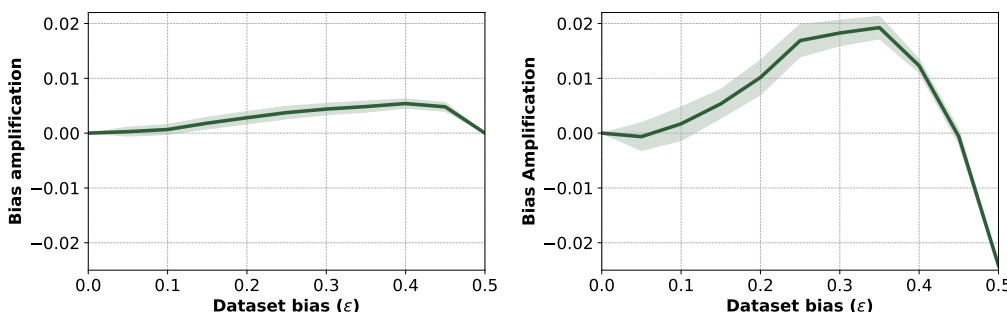

Figure 1: Bias amplification, $\text{BiasAmp}_{A \to T}$, as a function of the degree of bias, $\epsilon$, for (**left**) ResNet-18 models trained on the Fashion MNIST dataset and (**right**) ResNet-110 models trained on the CIFAR-100 dataset. Shaded regions indicate the 95% confidence intervals over 20 independent experiments.

**Bias amplification measure.** We adopt the directional bias amplification measure $\text{BiasAmp}_{A \to T}$ from [34]. This measure disambiguates different types of bias amplification and accounts for varying base rates of group membership.

The value of $\text{BiasAmp}_{A \to T}$ is 0 if the model predictions are exactly as biased as the labels in the dataset. If $\text{BiasAmp}_{A \to T}$ is negative, the model predictions dampen the bias present in the dataset and a positive $\text{BiasAmp}_{A \to T}$ value indicates that the model predictions amplify the bias in the dataset. Refer to Appendix A.1.2 for more details about the metric.

## 2.2 Results

We present the results of our experiments organized by research question (RQ). In accompanying materials, we also discuss the effect of training dataset size on bias amplification (Appendix A.3) and the relationship between overconfidence and bias amplification (Appendix A.4).

**RQ1: How does bias amplification vary as the bias in the data varies?** We perform experiments in which we vary the amount of bias in the Fashion MNIST dataset by generating training and test sets with different levels of bias, *i.e.*, by varying $\epsilon$.

The results from this experiment (left pane of Figure 1) show that when the training set is unbiased ($\epsilon = 0$), no bias amplification occurs because no bias is present. No bias amplification occurs when the training set is fully biased ($\epsilon = 1/2$), as it is impossible to amplify an already maximum bias. However, for intermediate $\epsilon$ values (*i.e.*, in partially biased training sets), the trained models amplify the bias present in the training data. Bias amplification generally *increases* with the amount of bias in training data, until the bias in the data is nearly maximized ($\epsilon = 0.5$).

We repeat the same experiment on the CIFAR-100 dataset with ResNet-110 models. The results (in the right pane of Figure 1) show a similar pattern. A notable difference, however, is that bias amplification is negative when the CIFAR-100 dataset is maximally biased ($\epsilon = 1/2$). We surmise that this happens because the group membership of CIFAR-100 images cannot always be recognized correctly by a model. To obtain zero bias amplification at $\epsilon = 1/2$, a model needs to be a perfect predictor of group membership. Hence, when the model incorrectly recognizes the group membership of some of the images, a negative bias amplification (*i.e.*, bias dampening) is obtained.

**RQ2: How does bias amplification vary as a function of model capacity?** It is well-known that the capacity of machine-learning models influences their classification performance. To understand how model capacity impacts bias amplification, we perform experiments in which we measure bias amplification while adjusting the capacity of our models. We adjust model capacity in three ways: (1) via the *depth* of the model; (2) via the *width* of the model; and (3) via the *regularization* of the model.

We focus on the CIFAR-100 dataset here because CIFAR-100 images are harder to classify than Fashion MNIST images: this makes it more likely that models with different capacities will produce substantially different predictions. We use the ResNet-110 model from our RQ1 experiments as our

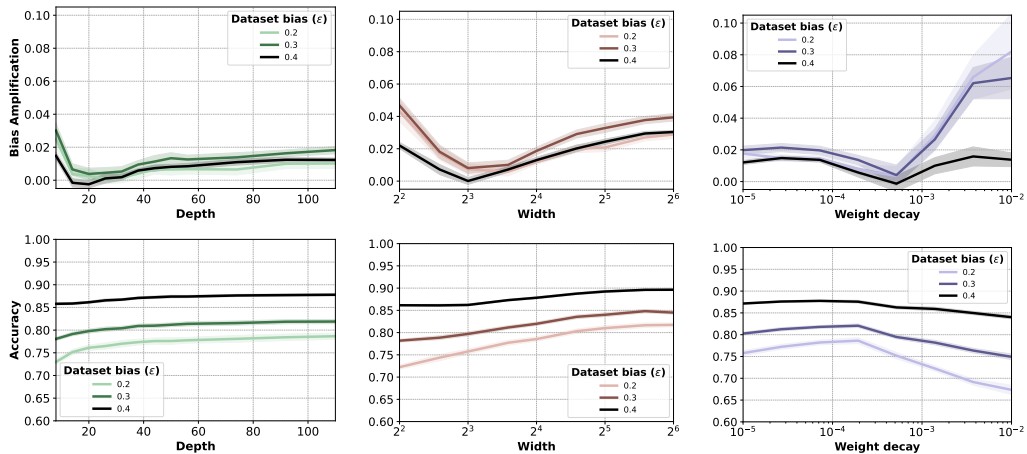

Figure 2: Bias amplification (**top**) and test accuracy (**bottom**) on the CIFAR-100 dataset as a function of three measures of model capacity. Each line represents a different amount of bias ($\epsilon$) in the training set. Shaded regions indicate the $95\%$ confidence intervals across 20 models. **Left:** Results for varying model depths. **Middle:** Results for varying model widths. **Right:** Results for varying weight decays.

base model. We experiment with depths that range between $8$ and $110$, widths ranging between $4$ and $64$, and logarithmically spaced weight decay values between $10^{-5}$ to $10^{-2}$. As before, we vary the dataset bias, $\epsilon$, between $0$ and $1/2$. The top row of Figure 2 shows the results of these experiments.

Irrespective of whether we vary model depth, width, or weight decay, the results suggest that bias amplification follows a "v-shape": it increases when model capacity increases beyond a certain level, but it also increases when model capacity is reduced below a certain level. We surmise there are different explanations for these two increases. When the capacity of a model is limited, it needs to rely on features that are easy to extract when making class predictions. When the dataset is biased ($\epsilon > 0$), the model thus relies on image inversion, which is easy to recognize, in its class predictions. This explains why bias amplification is relatively large when the model has low capacity.[2] In contrast, when the capacity of a model is large, bias amplification may increase because the model has the capacity to extract both features that indicate class membership and features that indicate group membership. This allows the model to use group membership features to increase the confidence of its predictions, which reduces the training loss.[3]

The relation between model capacity and bias amplification resembles the well-known relation between model capacity and *generalization error*. Models with insufficient capacity have high generalization error because they cannot model the data distribution well, whereas high-capacity models may have high generalization error due to *overfitting*. Our results suggests that there exists a model-capacity "sweet spot" in which bias is minimally amplified, akin to model-capacity sweet spot that minimizes generalization error (for a given training set).

To investigate whether the optimal model capacities for bias amplification and generalization error coincide, we plot the test accuracy of our models in the bottom row of Figure 2. Test accuracy increases monotonically with model depth and width, suggesting that the (overall) optimal model is larger than the range of models we experimented with. However, we do observe that a weight decay of $1.9 \cdot 10^{-4}$ appears optimal for test accuracy. This weight-decay value is smaller than the value that minimizes bias amplification ($5.2 \cdot 10^{-4}$), which suggests that model designers may sometimes have to trade off bias amplification and accuracy when tuning hyperparameters.

**RQ3: How does bias amplification vary during model training?** Thus far, we have only measured bias amplification of models that were trained until convergence for $500$ epochs. While it is

---

[2]This explanation relies on the assumption that *group membership* features are relatively easy to extract and, hence, that our observations may change had we not used image inversion to construct our synthetic groups. We investigate how the difficulty of recognizing group membership influences bias amplification in RQ5.

[3]Indeed, we surmise the increase in bias amplification in very high-capacity models is related to the tendency of such models to be overconfident [14]; we investigate this relation further in Appendix A.4.

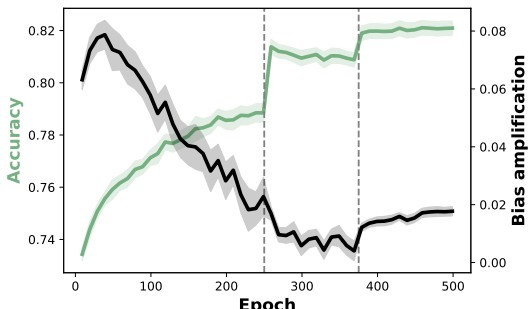 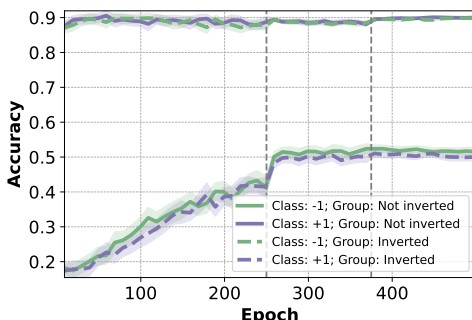

Figure 3: **Left:** Bias amplification and accuracy of ResNet-110 models during training on the CIFAR-100 dataset with a bias of $\epsilon = 0.3$. **Right:** Accuracy of the same models' per class-group combination. Shaded regions indicate the 95% confidence intervals across 50 models. Vertical dashed lines indicate epochs at which the learning rate of the mini-batch SGD optimizer is decreased.

feasible to train models until convergence on small datasets, it may not be practical to do so on very large training sets. To evaluate thow much bias amplification of a model varies during training, we measure bias amplification during the training of ResNet-110 models on a version of the CIFAR-100 dataset with bias $\epsilon = 0.3$.

The left pane of Figure 3 plots bias amplification and accuracy as a function of training epoch in this setting. The results in the figure show that bias amplification *varies greatly* during training; models amplify biases much more strongly in the early stages of training. Bias amplification gradually declines as training proceeds and the recognition accuracy of the model increases. However, the bias amplification increases again slightly in the final stages of training, in particular, after the learning rate is decreased to its smallest value. Notably, bias amplification appears to increase slightly every time the learning rate is decreased.

To better understand what drives these changes in bias amplification during training, we disaggregate the model's test accuracy into the four group-task combinations in the right pane of Figure 3. The model very quickly achieves high accuracy on examples for which the class label, $\{-1, +1\}$, matches the corresponding majority group, {inverted, not inverted}, per the bias in the dataset. By contrast, the accuracy on examples for which the class label does not match the majority group is very low in the initial stages of learning and increases much more gradually during training. We surmise this happens because group membership (image inversion) is easier to recognize than class membership (CIFAR-100 binary label). In the early stages of training, the model rapidly picks up on the easy-to-detect group membership signal as it provides the fastest way to reduce the model's loss. In turn, this leads to bias amplification because the model makes predictions based on group membership signals whilst ignoring class membership signals. As training progresses, the group membership signal loses value because it is not a perfect predictor of class membership (note that $\epsilon = 0.3$). Hence, the model starts to utilize more class membership signals as training progresses, which results in an increase in accuracy and a decrease in bias amplification.

To test this hypothesis, we perform an experiment in which we swap the role of the group and the class: *i.e.*, the class label now represents whether or not the image is inverted and the group label depends on the object depicted in the CIFAR-100 image. Indeed, we find that bias is dampened in the early stages of training as the model latches onto the easy-to-extract class membership signal first, but this largely disappears in the later stages of training as the model starts to utilize group membership signals for recognition as well. See Appendix A.2 for more.

**RQ4: How does bias amplification vary as a function of the relative difficulty of recognizing class membership versus recognizing group membership?** Hitherto, we repeatedly observed that bias amplification may depend on the relative difficulty of recognizing class membership versus recognizing group membership: as the group signal is easier to extract in our setup, models amplify bias more in early stages of training and/or when they have lower capacity. We perform a more detailed study of this relationship.

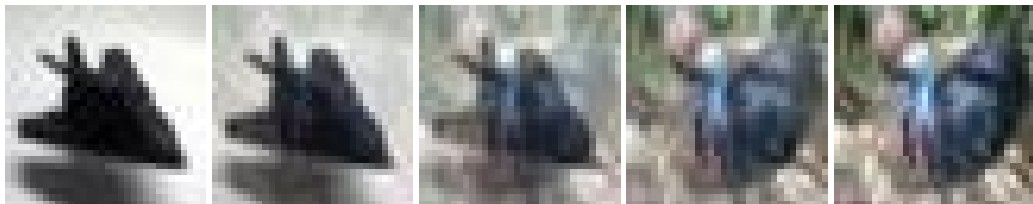

Figure 4: Example of different amounts of overlay ($\eta = 0.0, 0.2, 0.5, 0.8, 1.0$) for an example belonging to the "airplane" class and "bird" group. Only class information is visible when $\eta = 0.0$ (**left**); only group information is visible when $\eta = 1.0$ (**right**).

We alter our problem setup such that we can control the relative difficulty of class recognition and group recognition. We abandon our image-inversion setup and, instead, create datasets that contain a convex combination of two CIFAR-10 images: a "group image" and a "class image". By changing the weight of the convex combination, we can make the group image or the class image more prominent in the resulting image, thereby altering the difficulty of recognizing the class and the group.

We create the two groups, $a$ and $b$, by randomly choosing two CIFAR-10 classes that we sample group images from. We also randomly choose two CIFAR-10 classes to form the binary classification task (*i.e.*, one class is the positive class and the other the negative class). Next, we create an example by sampling a class image, $\mathbf{I}_{\text{class}}$, from one of the two classes and a corresponding group image, $\mathbf{I}_{\text{group}}$, from one of the two groups. We linearly mix these two images:

$$\mathbf{I} = \eta \mathbf{I}_{\text{group}} + (1 - \eta)\mathbf{I}_{\text{class}}, \tag{1}$$

where $\eta \in [0, 1]$ is a mixing parameter and the final example $\mathbf{I}$ is assigned the label of $\mathbf{I}_{\text{class}}$. Figure 4 shows an example of the resulting examples for different $\eta$ values. As before, we assign positive examples to group $a$ with probability $0.5 + \epsilon$ or to group $b$ with probability $0.5 - \epsilon$. Negative examples are assigned group $b$ with probability $0.5 + \epsilon$, and to group $a$ with probability $0.5 - \epsilon$.

When $\eta = 0$, this task reduces to classifying two classes from the standard CIFAR-10 images as the model cannot observe the group image at all. Conversely, directly recognizing class membership is impossible when $\eta = 1$ but recognizing group membership is easy in that setting. Hence, $\eta$ provides a knob that facilitates varying the relative difficulty of recognizing group membership versus class membership. Additionally, this new combinatorial method provides insight into bias amplification when there are specific visual features associated with individual sub-groups.

Figure 5 presents results of experiments in which we measure bias amplification as a function of the trade-off parameter, $\eta$, for different degrees of bias, $\epsilon$. It shows that bias is dampened when it is relatively difficult to recognize group membership (*i.e.*, when $\eta$ is low). When $\eta$ increases past the point where group information is more visible than class information ($\eta = 0.5$), however, the bias amplification starts to progressively increase and becomes positive for larger $\eta$. This observation provides additional evidence for the hypothesis that bias amplification depends heavily on the relative difficulty of recognizing group membership versus class membership.

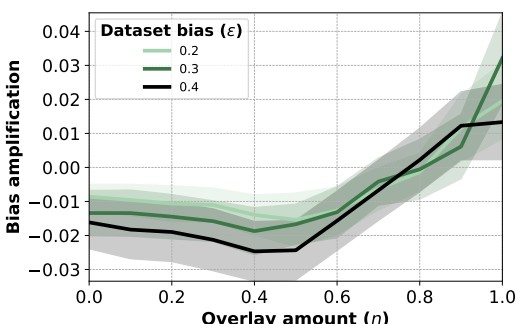

Figure 5: Bias amplification as a function of the relative difficulty of predicting class and group membership, $\eta$, for three different levels of bias, $\epsilon$. Recognizing class membership is easier for small $\eta$ values; recognizing group membership is easier for large $\eta$ values. Shaded regions indicate 95% confidence intervals across 20 models.

## 3 Related work

This study is one of many studies on fairness and bias amplification in machine-learning models.

**Fairness.** Prior work has introduced a large number of formulations of fairness, including equalized odds and equalized opportunity [15], fairness through awareness [10] or unawareness [13, 22], treatment equality [1], and demographic parity [10, 22]. Measures associated with these fairness formulations include differences in accuracy [1], differences in true or false positive rate [7, 15], and the average per-class accuracy across subgroups [5]. These measures differ from bias amplification measures in that they focus on correlations in the model predictions, whereas bias amplification focuses on *differences* between the correlations in the training data and those in the model predictions. In other words, bias-amplification measures discern between bias that is adopted from the training data and bias that is amplified by the model; fairness measures make no such distinction.

**Bias amplification.** The study of bias amplification is of interest because it allows us to study how design choices in our models, training algorithms, *etc.* contribute to bias in machine-learning models beyond biases in the training data [17]. Prior work has measured bias amplification using generative adversarial networks [6, 18], by considering binary classifications without attributes [24], and by measuring correlations in model predictions [19, 38]. In our work, we use the $\text{BiasAmp}_{A \to T}$ measure from [34], which addressed shortcomings in prior work [38], to measure bias amplification. Bias amplification has also been studied in the context of causal statistics [2, 26, 29, 30, 35], but that line of work has remained disparate from the study of bias amplification in machine learning. Despite the plethora of prior work on measuring bias amplification, little is known on *when and how* bias amplification arises in machine-learning models supporting vision tasks. Our study is among the first to shed some light on the context under which bias amplification occurs.

# 4 Discussion

The results of our experiments shed light on the conditions under which bias amplification can occur in machine-learning models for vision tasks. In particular, we find that bias amplification varies as a function of bias in the dataset, model capacity, training time, and the amount of training data. We also find that bias amplification depends on the relative difficulty of recognizing class membership and recognizing group membership. This creates a predicament as the Bayes error of those two recognition tasks are generally beyond the control of the model developer. Moreover, the model developer may not always be able to measure the difficulty of recognizing group membership empirically as doing so may involve developing a model that predicts sensitive attributes—something that model developers may want to avoid [20, 23, 31, 34].

Although our study does not resolve this predicament, it may provide some useful best practices to mitigate bias amplification as much as possible during model development. Our result suggests that there is value in using cross-validation to carefully select a model architecture, regularizer, and training recipe that minimizes bias amplification. Model developers may reduce bias amplification using the same tuning process that they routinely use to minimize classification error. Our study provides intuitions for how key levers available to the model developer can affect bias amplification. However such tuning does require access to sensitive attribute values, *viz.* group-membership information, and our study does not provide a complete overview of how all relevant levers influence bias amplification. We intend to perform a more comprehensive investigation in future work.

**Limitations.** While our study provides useful insights and suggests best practices, it also suffers from several key limitations. It is limited to binary classification tasks in the image-recognition domain and uses synthetic indicators of group membership. Further work is needed to understand how our findings apply to real world groups and biases, and how bias amplification manifests in different vision tasks and other modalities. We note that in recommendation tasks especially, bias amplification may arise in more complex ways because such systems generally have a human-in-the-loop influencing the behavior of the system [3].

Another limitation of our study is that it only studies bias *amplification*, which requires tradeoffs between other fairness guarantees and performance measures. Eliminating undesired biases altogether and ensuring fair, optimal performance thus requires careful design of the entire pipeline from data collection to model deployment.

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

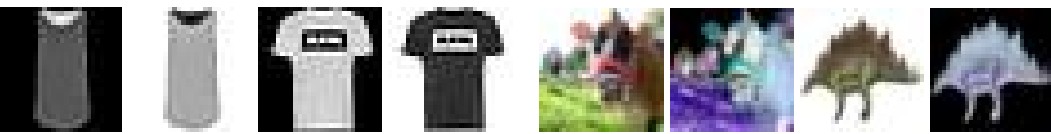

Figure 6: **Left:** Two examples of inversions performed on Fashion MNIST images. **Right:** Two examples of inversions performed on CIFAR-100 images. For each pair, the original image is on the left and the inverted image is on the right.

## A    Appendix

### A.1    Experimental set-up

Here we discuss additional details in our experimental set-up.

#### A.1.1    Model training.

All our models are residual networks [16] that are trained to minimize the binary cross-entropy loss between the model prediction and the true (binary) class label. We follow the training procedures in [16] and train our models using mini-batch stochastic gradient descent (SGD) with a Nesterov momentum [28] of 0.9 for 500 epochs. The models are trained using weight decay ($\ell_2$-regularization) with a decay parameter of $10^{-4}$. We warm up the training by setting the learning rate to 0.01 for one epoch as in [12]. Subsequently, the learning rate is set to 0.1 and decayed twice by a factor of 10 after 250 and 375 epochs. We train on a single GPU using a batch size of 128.

During training, we adopt the data augmentation procedure of [16] by: randomly cropping training images, flipping the resulting image horizontally with probability $1/2$, and resizing the crops to size $28 \times 28$ pixels (for Fashion MNIST) or $32 \times 32$ pixels (for CIFAR-10 and CIFAR-100). No data augmentation is used at test time. We normalize all images by subtracting a per-channel mean value and dividing by a per-channel standard deviation. When training models on CIFAR-10 and CIFAR-100, we follow [16] and pad the images with zeros.

#### A.1.2    Directional bias amplification.

We give a concise treatise of the measure here and refer the reader to [34] for further details.

Suppose we have a set of *classes*, $\mathcal{T}$, and a set of *groups*, $\mathcal{A}$. In our setup, $\mathcal{T} = \{-1, +1\}$ and $\mathcal{A} = \{\text{inverted}, \text{not inverted}\}$, where the binary labels $t \in \mathcal{T}$ were obtained by the random class assignment described above. The BiasAmp$_{A \to T}$ measure defines *bias* as a difference in the prevalence of a class label $t \in \mathcal{T}$ between groups $a \in \mathcal{A}$. For example, bias is present if inverted images are more likely to be positively labeled. Denote by $Pr(T_t = 1)$ the probability that an example in the dataset has class label $t$, and by $Pr(\hat{T}_t = 1)$ the probability that an example in the dataset is labeled as class $t$ by the model. With these definitions, [34] defines *bias amplification* as the difference in bias between the labels in the dataset and the labels predicted by the model:

$$\text{BiasAmp}_{A \to T} = \frac{1}{|\mathcal{A}||\mathcal{T}|} \sum_{a \in \mathcal{A}, t \in \mathcal{T}} y_{at}\Delta_{at} - (1 - y_{at})\Delta_{at}. \tag{2}$$

$\Delta_{at}$ measures the difference between the bias in the dataset and in the model predictions:

$$\Delta_{at} = Pr(\hat{T}_t = 1 | A_a = 1) - Pr(T_t = 1 | A_a = 1). \tag{3}$$

In the definition of BiasAmp$_{A \to T}$, $y_{at}$ alters the sign of the difference $\Delta_{at}$ to correct for the fact that the bias can have two directions. Specifically, $y_{at} \in \{0, 1\}$ is a binary variable that indicates the direction of the bias:

$$y_{at} = [Pr(T_t = 1, A_a = 1) > Pr(T_t = 1)Pr(A_a = 1)], \tag{4}$$

where $[\ldots]$ are Iverson brackets. In all our experiments, we compute BiasAmp$_{A \to T}$ by measuring both $Pr(\hat{T}_t = 1)$ and $Pr(T_t = 1)$ on the test set after training the model on the training set. The train and test datasets come from the same distribution.

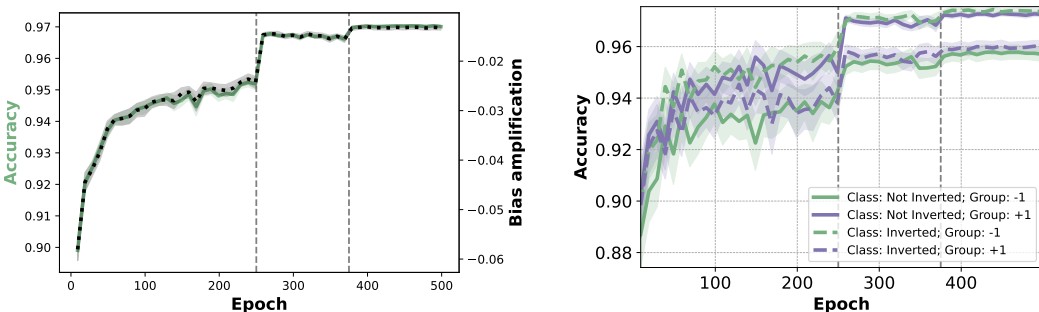

Figure 7: **Left:** Bias amplification and accuracy of ResNet-110 models during training on the CIFAR-100 dataset with a bias of $\epsilon = 0.3$ in which *the role of classes and groups is swapped* compared to the experiment in Figure 3: the class label indicates whether or not an image is inverted, and the group label is determined based on the visual content of the image. **Right:** Accuracy of the models' per class-group combination. Shaded regions indicate the 95% confidence intervals across 50 models. Vertical dashed lines indicate epochs at which the learning rate of the mini-batch SGD optimizer is decreased.

## A.2 Swapping group- and task-classes

In RQ4, we hypothesize that the early-stage bias amplification is due to group membership being easier to recognize than class membership in our setup. To test this hypothesis, we perform an experiment in which we swap the role of the group and the class: *i.e.*, the class label now represents whether or not the image is inverted and the group label depends on the object depicted in the CIFAR-100 image. We would expect the differences in accuracy between the majority / minority groups to disappear and bias amplification to actually be negative early on in training. As before, we measure bias amplification during training and plot the results in the left pane of Figure 7. The corresponding disaggregated accuracies are in the right pane of Figure 7. Indeed, we find that bias is dampened in the early stages of training as the model latches onto the easy-to-extract class membership signal first, but this largely disappears in the later stages of training as the model starts to utilize group membership signals for recognition as well.

## A.3 Effect of training size on bias amplification.

It is well-established that the error of machine-learning models can be reduced by increasing the amount of training data (as it reduces the estimation error [4, 33]). This raises the obvious question if bias amplification varies with training set size as well. To answer this question, we perform experiments in which we train ResNet-110 models on stratified subsamples of the CIFAR-100 training set. We vary the size of the subsamples to be a proportion, $p \in [0.1, 1.0]$, of the original training set. We increase the number of training epochs by a factor of $1/p$ so that each model performs the same number of parameter updates during training. We do not alter any of the other hyperparameters.

Figure 8 shows the results of our experiments. Whereas model accuracy increases monotonically with training set size, bias amplification varies in a more complex way. Beyond a certain training set size, bias amplification decreases with more training data. This is unsurprising: the additional training examples enable more accurate modeling of the data distribution, reducing bias amplification. However, bias amplification is also reduced when the training set becomes very small. We surmise this observation is due to overfitting: when trained on a small dataset, models tend to learn spurious correlations in that dataset rather than true statistical patterns such as the biases that exist in our training sets. The model cannot amplify bias if it is unable to capture that bias in the first place.

## A.4 Overconfidence and bias amplification

Our observation that models with higher capacity amplify bias more is reminiscent of observations that higher-capacity models tend to be more miscalibrated [14]. If high-capacity models are not explicitly calibrated, they are often overconfident in the sense that the accuracy of predictions that

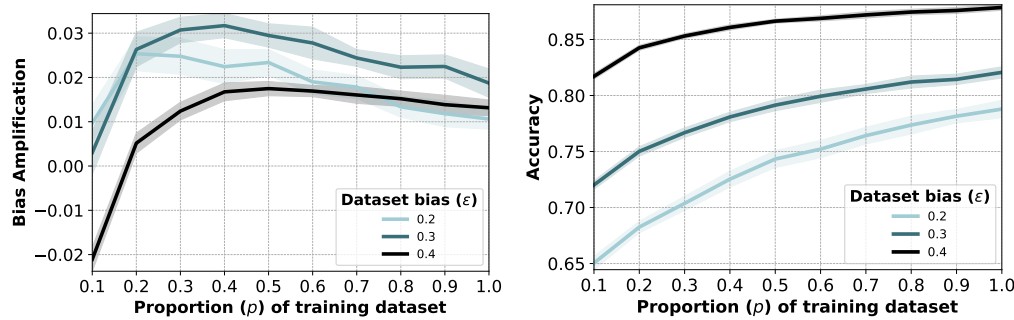

Figure 8: Bias amplification (**left**) and test accuracy (**right**) of ResNet-110 models on the CIFAR-100 dataset as a function of the proportion of the training set used for training the models. The number of epochs for each model is scaled depending on the amount of training data used. Shaded regions indicate the 95% confidence intervals across 20 models.

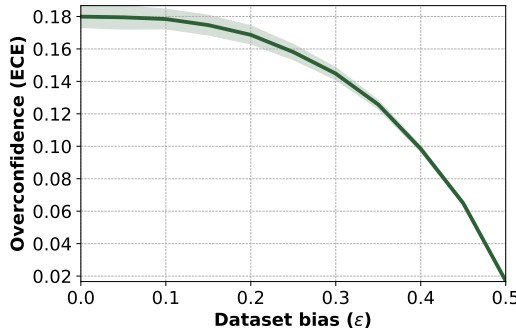

Figure 9: Expected calibration error (ECE) of ResNet-110 models on the CIFAR-100 dataset as a function of dataset bias, $\epsilon$. Shaded regions indicate the 95% confidence intervals across 20 models.

they make with, say, 90% confidence is lower than 90%. We perform experiments to investigate if bias amplification is correlated to such model overconfidence.

To do so, we measure the overconfidence of our models in terms of the expected calibration error (ECE) [27]. The ECE measures the expected value of the (absolute) difference between the model accuracy and the model confidence:

$$ECE(\hat{P}) = \mathbb{E}\left[\left|Pr(\hat{Y} = y|\hat{C} = c) - c\right|\right], \tag{5}$$

where $\hat{Y}$ and $\hat{C}$ are random variables indicating the class label of an example and the model-prediction confidence for that same example, respectively, and the expectation is over all possible confidence values $c \in [0, 1]$. Because we only have access to a finite number of samples of the distribution $p(\hat{C})$, we approximate the expected value by binning $p(\hat{C})$ into 15 values and averaging those values, weighted by the number of examples per bin. A higher ECE value indicates a larger discrepancy between the prediction confidence values and the corresponding accuracies, *i.e.*, a higher degree of model overconfidence.

Figure 9 shows ECE as a function of the bias in the dataset, $\epsilon$, for ResNet-110 models on CIFAR-100. We observe that model overconfidence decreases with bias in our experiment, because the task becomes easier as bias increases: if a task is very easy, a model is generally less overconfident as it correctly predicts nearly every example.

Next, we study the relation between overconfidence and bias amplification by varying the capacity of the model. Figure 10 shows this relation for three levels of dataset bias, $\epsilon$, and for three model-capacity measures: depth, width, and weight decay. Darker points in the figure correspond to higher-capacity models. The results show that bias amplification initially decreases as model overconfidence increases

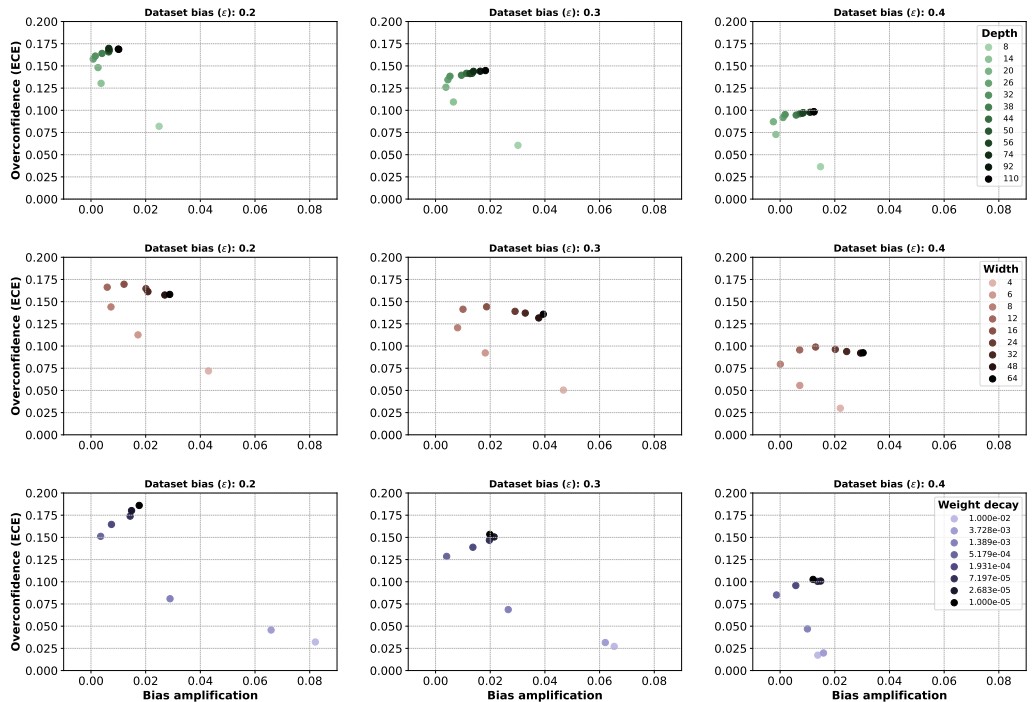

Figure 10: Bias amplification and expected calibration error (ECE) of ResNet models of varying depth (**first row**), width (**second row**), and weight decay (**third row**) on the CIFAR-100 dataset, for three values of the dataset bias, $\epsilon$. Results are averaged over 20 runs.

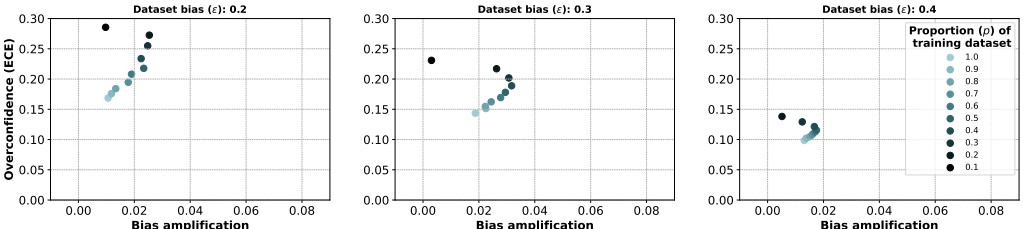

Figure 11: Bias amplification and expected calibration error (ECE) of ResNet models of varying training dataset size on the CIFAR-100 dataset, for three values of the dataset bias, $\epsilon$. Results are averaged over 20 runs.

(for low-capacity models), but that bias amplification and overconfidence both increase for higher-capacity models.

Finally, Figure 11 studies the relationship between bias amplification and overconfidence as the size of the training set changes. Darker points in the figure correspond to smaller training sets. As expected, reducing the number of training examples increases the model's overconfidence (ECE). Bias amplification, however, initially increases as the training set size decreases but decreases again when the training set becomes very small.

