# OpenReview forum: "Bias Amplification in Image Classification"
_NeurIPS.cc/2022/Workshop/TSRML — TSRML2022_

### Official Review · Reviewer_q1m8 · 2022-10-17
**A interesting and high-quality paper on bias amplification in neural networks**

**Overall Rating:** 7

**Summary:**

The paper studies bias amplification in machine learning models. The study is performed on a simple image classification problems which facilitate the introduction of controlled synthetic biases. The paper targets four research questions: how does bias amplification vary as (i) the bias in the data varies, (ii) as a function of model capacity, (iii) during model training, and (iv) as a function of the relative difficulty of recognizing class membership versus recognizing group membership.

**Strengths:**

I am not an expert on biases in neural networks, and the following comments should be read in that context.

1. The topic of bias amplification is of high interest to the field.

2. The paper is very well-written and easy to follow.

3. The research questions are well-defined and of high interest to the field.

4. The empirical evaluation is thorough, includes relevant statistical measures such as confidence intervals, and addresses the relevant research questions well.

**Weaknesses:**

1. The paper mainly considers synthetic biases based on inverting the images. This is a relatively simple feature for neural networks to detect, and the discussion on how more subtle biases would affect the results in the experiments related to research questions 1-3 is limited. While question 4 addresses this issue to some degree, the authors could discuss in more detail how the results from these experiments affect the conclusions drawn from the experiments related to research questions 1-3.

Minor:

2. On line 114, I assume "width" is the number of channels in the convolutional layers; however, a clarification would help.

**Overall Recommendation:**

The paper is of high quality, and the few weaknesses are outweighed by the strengths of the paper.

**Review Confidence:**

3: The reviewer is fairly confident that the evaluation is correct

---

### Official Review · Reviewer_ophC · 2022-10-19
**Interesting and novel analysis of bias amplification**

**Overall Rating:** 8

**Summary:**

This paper analyzes the bias amplification effect whereby existing biases in the training data are not only replicated by a trained deep neural network but are amplified. The paper proposes a simple and interesting bias mechanism, which is also controllable. A dataset is first split into two classes then with a rate 1/2-\epsilon for one class and 1/2+\epsilon for the other, random samples are inverted. Thus the dataset is biased in a controllable manner, and for \epsilon =1/2 class membership is completely defined by the bias. The authors then ask

1) How does bias amplification vary as the bias in the data varies?
2) How does bias amplification vary as a function of model capacity?
3) How does bias amplification vary during model training?
4) How does bias amplification vary as a function of the relative difficulty of recognizing class membership versus recognizing group membership?

Through experiments on the FashionMnist, Cifar-10 and Cifar-100 datasets using realistic architectures the authors show that

1) Moderate bias results in bias amplification.
2) Constrained model capacity results in bias amplification. The optimal capacity for bias amplification to *not* exist is different for the optimal capacity for generalization.
3) Bias amplification first decreases and then increases with model training, even if training accuracy hasn't plateaued.
4) Roughly: More difficult datasets result in more bias amplification.



**Strengths:**

The submission is clearly written and is (to the best of my knowledge) novel and significant. That neural networks reproduce bias that exists in training data is a well-known and very relevant issue for the real-life adoption of neural networks. The observation that bias might also increase specifically because of using a machine learning approach for prediction is in my opinion very timely and relevant for discussions on when and how deep learning approaches should be used for critical applications.

**Weaknesses:**

I don't see any weaknesses in this submission which I think is of a very good level for a workshop track.

**Overall Recommendation:**

I recommend that this submission be accepted, due to its novelty, clarity, and relevance. However, I am not an expert on the topic and have set my confidence level accordingly.

**Review Confidence:**

3: The reviewer is fairly confident that the evaluation is correct

---

### Official Review · Reviewer_wUcJ · 2022-10-22

**Overall Recommendation:** Given the interesting experimental se…
**Overall Rating:** 8

**Summary:**

This paper studies the bias amplification in machine-learning models for vision tasks. The authors perform a carefully controlled experimental study on different conditions that could lead to bias amplification, including model accuracy, model capacity, and amount of training data.

**Strengths:**

1. The controlled experiments provide fine-grained findings/insights for understanding bias amplification.

2. This paper identifies an interesting finding in bias amplification, the difficulty of *the classification task relative to the difficulty of recognizing group membership plays an important role* in the bias amplification.

**Weaknesses:**

N/A

**Review Confidence:**

4: The reviewer is confident but not absolutely certain that the evaluation is correct

---

### Decision · Program_Chairs · 2022-10-23

Accept